# Malnutrition in Pediatric Chronic Cholestatic Disease: An Up-to-Date Overview

**DOI:** 10.3390/nu13082785

**Published:** 2021-08-13

**Authors:** Maria Tessitore, Eduardo Sorrentino, Giuseppe Schiano Di Cola, Angelo Colucci, Pietro Vajro, Claudia Mandato

**Affiliations:** 1Department of Medicine, Surgery and Dentistry “Scuola Medica Salernitana”, Chair of Pediatrics and Residency Program of Pediatrics, Via S. Allende, University of Salerno, 84081 Baronissi, SA, Italy; mariatessitore20@gmail.com (M.T.); eduardosorrentino30@gmail.com (E.S.); schianodicolag@gmail.com (G.S.D.C.); angelocolucci2@gmail.com (A.C.); pvajro@unisa.it (P.V.); 2Department of Pediatrics, Santobono-Pausilipon Children’s Hospital Via M. Fiore, 80129 Naples, Italy

**Keywords:** cholestasis, chronic liver diseases, malnutrition, nutritional needs, pediatrics

## Abstract

Despite recent advances, the causes of and effective therapies for pediatric chronic cholestatic diseases remain elusive, and many patients progress to liver failure and need liver transplantation. Malnutrition is a common complication in these patients and is a well-recognized, tremendous challenge for the clinician. We undertook a narrative review of both recent and relevant older literature, published during the last 20 years, for studies linking nutrition to pediatric chronic cholestasis. The collected data confirm that malnutrition and failure to thrive are associated with increased risks of morbidity and mortality, and they also affect the outcomes of liver transplantation, including long-term survival. Malnutrition in children with chronic liver disease is multifactorial and with multiple potential nutritional deficiencies. To improve life expectancy and the quality of life, patients require careful assessments and appropriate management of their nutritional statuses by multidisciplinary teams, which can identify and/or prevent specific deficiencies and initiate appropriate interventions. Solutions available for the clinical management of these children in general, as well as those directed to specific etiologies, are summarized. We particularly focus on fat-soluble vitamin deficiency and malnutrition due to fat malabsorption. Supplemental feeding, including medium-chain triglycerides, essential fatty acids, branched-chain amino acids, and the extra calories needed to overcome the consequences of anorexia and high energy requirements, is reviewed. Future studies should address the need for further improving commercially available and nutritionally complete infant milk formulae for the dietary management of this fragile category of patients. The aid of a specialist dietitian, educational training regarding nutritional guidelines for stakeholders, and improving family nutritional health literacy appear essential.

## 1. Introduction

Cholestasis is regarded as reduced bile formation or flow, leading to a decreased concentration of bile acids in the intestine, and the retention within the blood and the liver of biliary substances which are normally excreted into bile [1]. In the pediatric population, cholestasis predominantly affects neonates and infants, around 1 in every 2500 term infants [2]. A substantial proportion have chronic courses of disease (chronic cholestatic liver diseases (CCLD)). Despite recent advances, effective therapies remain elusive, so that many conditions progress to liver failure and may necessitate early liver transplantation. The most common causes of prevalently extrahepatic (surgical) cholestatic jaundice are biliary atresia (BA) (25–40%) and choledochal malformations. As shown in Figure 1, the etiologies of intrahepatic (medical) disorders are more numerous and include a mounting group of genetic disorders related to bile acid transport or synthesis, inborn errors of protein and carbohydrate metabolism, syndromes, mitochondrial and endocrine diseases, hepatic infections, and parenteral nutrition-associated cholestasis (PNAC)) [3,4,5].

Malnutrition is a common complication of CCLD, which may increase morbidity and mortality at all ages, particularly in pediatric patients due to their specific developmental aspects. The proportion of malnutrition due to intensely reduced bile acid-dependent absorption of fats and fat-soluble nutrients is, however, difficult to quantify compared to that caused by fibrosis/cirrhosis per se.

Chronic enteropathy secondary to associated portal hypertension, poor nutrient intake, increased energy needs, and endocrine dysfunction are further mechanisms [6,7]. Pediatricians and dietitians should be specifically experienced in pediatric CCLD. Data show that improved nutritional consultations after stakeholders’ educational training on nutritional guidelines were associated with lower readmissions of adult patients [8]. Similarly, parental health illiteracy necessitates appropriate interventions targeting social and cultural family circumstances [9].

In this context, we performed a comprehensive review of the literature in the PubMed and Google Scholar databases for studies published during the past 20 years linking nutrition to pediatric chronic cholestatic liver disease. This search was completed by personal knowledge, a snowball strategy by searching for any relevant previous studies within the list of references of analyzed articles and/or citation tracking by a computer aided manual search (Figure 2). Here, we report and put together the multiple pieces of the malnutrition puzzle in children and summarize solutions that are available in everyday clinical practice.

## 2. Causes of Malnutrition in Cholestatic Children

As outlined earlier, malnutrition in CCLD depends on several cooperating factors (Figure 3).

### 2.1. Metabolic Changes

The liver plays an important role in the regulation of nutrient metabolism. Therefore, when the liver is damaged by any type of injury, nutrient digestion, absorption, storage, and utilization are affected. The decrease in hepatic and muscular glycogen reserves results in the activation of alternative metabolic pathways with an increased release of amino acids (and hyperammonemia), increased fat oxidation, and rapid use of fat, resulting in hypercholesterolemia and low triglyceride levels. The associated cholestatic component results in further specific problems.

### 2.2. Poor Nutrient Intake

The reduced dietary intake in cholestatic children may result from anorexia, nausea, vomiting, changes in taste perception, and early satiety [10].

Anorexia may presumably be the result of a change in amino acid metabolism with increased tryptophan levels and ensuing increases in brain serotonergic activity, which are reportedly involved in the regulation of eating behavior [11]. Nausea and vomiting are triggered by increased pro-inflammatory cytokines [12], organomegaly, and ascites, resulting in reduced gastric capacity and consequent early satiety [13].

Chronic pruritus, due to accumulation of bile acid (BA) and other pruritogenic substances, may be particularly severe and disturbing especially in PFIC and Alagille syndrome. It may dramatically reduce patients’ quality of life and therefore warrants serious and prompt attention with adequate treatment [1,14,15,16,17,18].

Finally, a deficiency in zinc or magnesium contributes to a change in taste perception, which may be further aggravated by the use of poorly palatable special formulas [8] and/or dietary modifications with sodium, fluid, or protein restrictions.

### 2.3. Increased Requirements or Malabsorption/Maldigestion of Multiple Nutrients

#### 2.3.1. Increased Energy Needs

Children with cholestatic disease have often (but not unanimously) [19] been reported to suffer from a *hypermetabolic state,* with increased energy expenditure, probably due to the intracellular activation of thyroid hormone by bile acids [20,21]. In biliary atresia, a 29% increase in energy requirements, compared to healthy controls, has been calculated [20]. In children with end-stage liver disease, the energy demand may further increase up to 150% of that of predicted normal for a given height and weight, especially in cases of complications such as episodes of sepsis from peritonitis, cholangitis, or variceal bleeding [10,22].

Because the equations commonly used to predict resting energy expenditure (cREE), i.e., the Food and Agriculture Organization/World Health Organization/United Nations University Schofield (weight, and weight and height) equations [23] (Table 1) perform inadequately, especially in the case of end-stage liver disease cholestasis, indirect calorimetry should be used when available to guide energy provisions, particularly in children who are already malnourished [24].

#### 2.3.2. Water and Electrolytes

The fluid and electrolytes requirements are normal for maintaining weight, unless restriction is needed because of ascites or edema. In infancy, a sodium intake of at least 1 mmol/kg/day and potassium intake of about 2 mmol/kg/day are usually appropriate [6].

Sodium should not be added to correct hyponatremia due to the systemic vasodilation and arterial underfilling in patients with advanced liver disease who have developed ascites.

#### 2.3.3. Carbohydrates

For CCLD patients, carbohydrates are a major source of energy (about two-thirds of non-protein energy) as they are better accepted than lipids or proteins due to a more pleasant taste [25]. They can be given as monomers, polymers, and starch. Short-chain polymers (maltodextrin) are generally used because of their low osmotic load, which prevents the onset of diarrhea. Starch is a possible alternative, but it can cause abdominal bloating and diarrhea because it has poor digestibility due to the enzymatic immaturity of amylases at an early age.

#### 2.3.4. Proteins

In patients with biliary atresia, increased oxidation of both exogenous and endogenous proteins has been shown, which results in zero nitrogen balance oxidation leading to muscle proteolysis [20,25,26]. Children with advanced CCLD require a higher protein intake for catch-up growth (a protein/energy ratio of 10%). Currently, the specific needs in cholestasis are not known; infants with severe cholestatic liver disease have a daily protein intake need of approximately 2–3 g/kg/day to ensure a positive nitrogen balance and proper growth [27]. In childhood, the requirement becomes progressively lower, except when complications (e.g., cholangitis) increase proteolysis and lead to negative protein balance [28].

In the absence of encephalopathy, hyperammonemia up to 120 mmol is well tolerated, without evident side-effects, and does not require protein restriction [29]. Protein restriction is rarely needed in children with cholestatic liver disease, unless there is a urea cycle disorder (UCD) related-cholestasis. Protein restriction can be considered if encephalopathy occurs as a result of liver failure and of portal hypertension-related portosystemic shunts. In this case, protein intake should be limited to 0.5–1.0 g/kg/day although restriction to <2 g/kg/day should not be continued in the long term, as it can induce endogenous muscle protein consumption. If unresponsive, patients may benefit from branched-chain amino acid (BCAA) supplementation [27].

Children with cholestatic liver disease have low levels of serum BCAAs and elevated ratios of aromatic amino acids to BCAAs, due to an alteration in amino acid kinetics and increased expenditure of BCAAs in the muscle [30]. In other words, low levels of BCAAs reflect increased BCAA utilization in muscle. Therefore, diets rich in BCAAs may have significant advantages. In experimental biliary atresia, a formula enriched with BCAAs improved weight gain, protein mass, muscle mass, nitrogen balance, body composition, and bone mineral density [31]. In addition, a diet enriched in BCAAs in children with end-stage liver disease resulted in an improvement in nutritional status and total body potassium [32]. To improve palatability without reducing energy intake, an ideally formulated product should preferably include whey protein (3 g/kg/day), at 2.6 g/100 mL in reconstituted formula, enriched in BCAAs to 10% [32,33,34,35].

#### 2.3.5. Lipids and Bile–Acid-Dependent Absorption of Fats and Fat-Soluble Nutrients

Lipids are the main source of energy supplied through breast milk. During the first years of life, lipids are necessary for growth, development, and providing essential polyunsaturated fatty acids (PUFAs) and fat-soluble vitamins [36]. Although lipids are less palatable than other macronutrients, they are an important supplement because of their high energy, low osmolarity, and content of essential PUFAs [23,37]. In cholestasis, the considerable decrease in long-chain triglyceride absorption due to impaired micellization requires balancing energy losses with extra energy supply. Despite the possible development of steatorrhea, animal studies suggest that overall nutritional status benefits from a high rather than a restricted fat intake [38]. Oral bile salt substitution therapy would be cumbersome and impractical. However, some cholestatic diseases are treated with ursodeoxycholic acid (UDCA) (20–30 mg/kg/day), as it increases bile formation and counteracts the hydrophobic effect of retained bile acids on cell membranes. However, it has no effect on micelle formation and lipid absorption.

#### 2.3.6. Medium Chain Triglycerides and Long-Chain Triglycerides

Although medium-chain triglycerides (MCTs; C-8 to C-12 fatty acids) have a lower energy content (about 16% lower than long-chain triglycerides LCTs), they are used as a lipid supplement (MCT-enriched formula) because their shorter chains allow them to spread passively (bound to albumin) through the gastrointestinal tract and be directly absorbed into the portal circulation. In fact, unlike LCTs, MCTs do not require micellar solubilization and re-esterification [39] because they completely bypass the lymphatic system [40], with approximately 95% bioavailability, even in very cholestatic children [41]. Unless the levels of MCT exceed the metabolic capacity of the liver, they undergo liver metabolism, with energy release. This happens independently of carnitine, which is required for the transport of long-chain fatty acids through the mitochondrial membrane. However, MCTs may reduce appetite, probably due to interaction with the peptides YY and cholecystokinin, with possible interference with the metabolism of adipose tissue [42].

In CCLD children, the ideal fat content and ratio of MCT to LCT are difficult to determine. The optimal proportion of total lipids as MCTs for nutritional management is between 30 and 50% [10,43]. Much higher MCT content in the diet (i.e., >80%) without adequate supplementation of PUFA should be avoided since it can lead to a deficiency in essential fatty acids [44]. Limited data have shown that infants fed with 30 or 70% MCT instead of a 50/50% mixture of MCT/LCT have better fat solubilization and growth [45,46]. For these reasons, most infant formulae with MCTs have an MCT/LCT ratio of about 1/1 and are supplemented with essential fatty acids. Lipid intake and the MCT ratio should be tailored for optimal weight gain and growth. [46].

#### 2.3.7. Essential Fatty Acids

Essential fatty acids (EFAs) are macronutrients that must necessarily be included in the diet because humans cannot synthesize their precursors. Linoleic acid (C18:2n–6) and linolenic acid (C18:3n–3) are the two main EFAs. They undergo hepatic and cerebral elongation into long-chain polyunsaturated fatty acids (LCPs, PUFAs, i.e., arachidonic acid (AA) and docosahexaenoic acid (DHA)). The latter are important for the growth and development of membrane-rich tissues such as the brain and retina [44]. PUFAs are also precursors of eicosanoids, which improve immune function, reduce systemic inflammation [47], and participate in platelet aggregation. These important biological roles make PUFA deficiency in cholestatic liver disease a concern. PUFA and EFA deficiency may derive from low intake, fat maldigestion/malabsorption, and inefficient elongation of EFA precursors secondary to dysfunctional hepatocytes and enhanced peroxidation of lipids [6]. It can also be iatrogenic, when diets are exceedingly high in MCT and low in LCT [48]. Clinically, PUFA and EFA deficiencies present with dry and rough skin, poor growth, numbness, paresthesia, and vision impairment. These clinical signs may go unrecognized or be misdiagnosed as vitamin deficiencies [49]. Testing for PUFA and EFA deficiency, which requires the total fatty acid profile in red blood cells, is costly and not commonly available. It is recommended every 3–6 months especially in the case of severe maldigestion/malabsorption or if the diet comprises exclusive MCT lipids (>80%), or lower in severe cholestasis [6]. Attention is required when linoleic acid, α-linolenic acid, eicosapentanoic acid, and/or docosahexaenoic acid levels are low and either clinical signs or severe deficiencies in fat-soluble vitamins are resistant to supplementation. Importantly, the classic marker of a ratio of triene/tetraene >0.2 is not a sufficient marker when testing for PUFA and EFA deficiency [50]. Unlike in breast milk, formulae contain little LCP [51], and this can lead to possible malnutrition unless egg yolks or fish oil are administered at weaning [22,51].

Estimating appropriate PUFA and LCP intakes for healthy infants and children is not easy and is even more difficult for children with cholestasis. Supplementation should exceed 10% of total energy, but even this may not be sufficient. In fact, when liver damage increases, hepatic PUFA conversion to LCPs is impaired, and LCP deficiency occurs even with an adequate EFA supply. In advanced CCLD, LCP deficiency is difficult to correct; it has been documented that even 1 year after liver transplantation, the LCP status may still not be entirely reversed [52].

That said, there are currently no studies showing the functional effects of LCP supplementation in cholestatic children based on which exact recommendations can be provided. In infants, breast milk can be supplemented with breast milk fortifiers. If breastfeeding is not possible, an MCT-enriched formula should be used. To better meet the energy needs of infants, another option is to increase the caloric density of the formula to 0.8–1 kcal/mL [53]. Older children can improve their EFA intake by adding canola oil, sunflower oil, soybean oil, fish oil, and egg yolks to their diet [22,54].

#### 2.3.8. Fat-Soluble Vitamins

In cholestasis, the reduced secretion of bile acids into the intestinal lumen also induces the malabsorption of fat-soluble vitamins (FSVs) (vitamins A, D, E, and K). This is more frequent when direct-reacting (i.e., conjugated) bilirubin serum levels are greater than 2 mg/dL [55]. In infants with biliary atresia, total serum bilirubin appears to be a better, although still imperfect, predictor of FSV deficiency [56]. The role of serum bile acid as a surrogate marker for guiding the monitoring of FSV deficiency in chronic intrahepatic cholestasis is still undefined.

When cholestasis begins in infancy, vitamin stores present at birth are rapidly depleted; this results in biochemical and clinical signs of fat-soluble vitamin deficiency as early as 4–12 months of age if supplementation has not begun [27]. The serum levels of vitamins and prothrombin time should be monitored to allow the proper adjustment of dosages to the patient’s specific needs and the prompt treatment of possible side effects. In particular, vitamin A levels must be closely monitored during supplementation because high levels may cause neurologic and hepatic damage [1].

Vitamin D insufficiency, defined as 25-OH vitamin D less than 30 ng/mL, is present in more than half of cholestatic patients, and this is positively correlated with serum calcium [57]. Decreased bone mineral density (BMD) was present in more than half of the studied cholestatic patients and was correlated with low serum calcium rather than vitamin D levels. Decreased BMD and dental disorders in cholestatic children are related to the level of hyperbilirubinemia [57]. Table 2 synoptically summarizes the effects of individual fat-soluble vitamin deficiencies and toxicities as reported in pediatric practice [25,27,33,57,58].

#### 2.3.9. Water-Soluble Vitamins and Minerals

Cholestatic children may also present with malabsorption of water-soluble vitamins and minerals. Therefore, it is necessary to supplement vitamins normally present in the diet, using standard pediatric multivitamins. The doses recommended are twice the RDAs [59].

The liver and gut microbiota play important roles in regulating most trace elements; therefore, the impairment of liver function and/or dysbiosis can negatively affect metabolism. In addition, the administration or depletion of these trace elements may also cause liver dysfunction [60] Zinc, selenium, calcium, phosphate, magnesium, and iron may be deficient as well, and they should be supplemented according to the plasma levels. Zinc and selenium are important antioxidants. Reduced zinc concentrations lead to poor linear growth, hypogeusia, anorexia, impaired immune function [42], skin rashes, and diarrhea [61,62]. Identifying zinc deficiency may be difficult because plasma zinc levels do not correlate with tissue zinc content; thus, clinicians should suspect zinc deficiency in patients on the basis of gastrointestinal and dermatological manifestations [63].

Calcium and phosphate deficiency may also develop during cholestasis, leading to bone abnormalities that are unresponsive to the normalization of vitamin D status [27]. Magnesium deficiency may occur as well, contributing to the metabolic bone disease of CCLD. Liver transplantation has favorable effects on osteopenia and vitamin D deficiency. It has been shown that in infants and children <2 years of age, the bone mineral content normalizes approximately 11 months after transplantation, provided there is a sufficient period of normal serum 25-OH-D levels [64].

Iron deficiency is uncommon in chronic liver disease. It may occur, however, in cases of reduced intake and chronic blood loss from gastrointestinal bleeds (esophageal varices or portal hypertensive gastropathy) [22]. As iron can promote oxidative stress, carcinogenesis, and fibrogenesis, it should not be supplemented on a regular basis [65]. On the contrary, chronic cholestasis may be accompanied by the excessive accumulation of copper in the liver [66] and possibly contribute to its further damage.

### 2.4. Endocrine Dysfunction

The liver is the main source of insulin-like growth factor (IGF-1) and its major circulating binding protein, IGF binding protein 3 (IGF-BP3). IGF-1 stimulates the anabolic actions of growth hormone (GH). In cholestatic liver diseases, IGF-1 and IGF-BP3 formation is reduced, resulting in an impaired GH/IGF-1 axis. IGF-1 levels further decrease due to GH resistance, caused by the downregulation of GH receptors [22]. This condition contributes to aggravating growth failure in these children, who may already have an underlying predisposition to short stature (e.g., Alagille Syndrome) [67].

## 3. Issues in the Nutritional Management of Children with Cholestasis

According to the above considerations, it is evident that CCLD negatively affects the nutritional status in infancy (i.e., when growth rates are the highest), thus compromising clinical outcomes for cholestatic children who have end-stage liver disease [68], and is present in about 80% of cases [69]. CCLD increases the mortality and morbidity associated with underlying diseases and significantly influences the outcomes of liver transplantation in children [70]. It is therefore necessary to ensure proper nutritional support to prevent further liver damage and improve the likelihood of successful liver transplantation [36]. All children with cholestatic liver disease should have a clinical nutritional evaluation with an intervention and follow-up plan reported in their care records, with a frequency appropriate for the patient’s clinical course. The evaluation of the nutritional status of these patients is based on anthropometric, biochemical, and instrumental indicators [71,72], as summarized below and in Figure 4.

### 3.1. Anthropometric Measurements

In children with cholestatic liver disease, the use of weight-for-age and weight-for-height measurements can be inaccurate for nutritional evaluation, owing to visceromegaly, subclinical edema, and/or ascites as a result of excessive tissue sequestration of water, which results from an abnormal intravascular colloid osmotic status, renal retention of salt and water, and hyperaldosteronism [1,73].

Stunting is determined by serial measurements of the height–age index (height-for-age ≤ −2 SD of the WHO child growth standards median). Length (for children < 2 years) or height (for children > 2 years) may be a good indicator of chronic malnutrition, while short-term changes in nutritional status need to be tracked using other parameters such as the mid upper arm circumference (MUAC) and triceps skin fold (TSF). These measures provide information regarding the patient’s body composition and are less affected by fluid overload or other complications of end-stage liver disease [10], and they are reported to decline before changes in weight or height become apparent [74].>

The MUAC captures both muscle and adipose tissue mass, and it is relatively stable in the first years of life. An absolute value < 12.5–13.0 cm [75] or a Z score < −2 [76] indicate moderate to severe malnutrition. Therefore, by using the MUAC as a screening tool, it is possible to quickly identify children with moderate to severe malnutrition, who are at increased risk for death and need nutrient supplementation and treatment for their underlying disease [36]. The TSF reflects adiposity and is a good indicator of energy reserve depletion during cholestatic liver disease [77]. It has been shown to be more sensitive for acute malnutrition and wasting than weight-for-height Z scores [78].

Globally, children with cholestatic liver disease have decreased MUAC, TSF, and mid-upper arm muscle area Z scores than healthy controls [79]. The frequency of the measurement is variable, depending on the grade of nutritional status, and can range from every 2 weeks to every 3 months [6].

### 3.2. Biochemical Markers

Serum protein levels (albumin, prealbumin, transferrin, and retinol-binding protein) alone cannot be used as an indicator of malnutrition, as their production is influenced by hepatic disease, sepsis, inflammation, and hydration status [80]. Albumin is not sensitive to acute changes in the nutritional statuses of children with chronic liver disease, as it has a long half-life (18–20 days). In addition, it may be depressed due to inflammation or acute physiologic stress [81]. Prealbumin is a more sensitive marker for the severity of malnutrition and/or adequacy of nutritional support [74]; it has a half-life of 2 days [15], low body reserves, responds quickly to nutritional status [80], and its production in the liver is maintained until late in liver disease [82]. Retinol-binding protein has a half-life of 12 h, which makes it the best indicator of recent dietary changes [80].

### 3.3. Other Investigations

The evaluation of nutritional status in cholestatic children also includes instrumental investigations that assess body composition: dual-energy X-ray absorptiometry (DXA), bioelectrical impedance (BIA), and indirect calorimetry. DXA and BIA provide a measure of fat and free-fat mass, which are helpful when designing a nutritional rehabilitation approach, although the accuracy of these tools can decrease due to fluid overload [83].

Indirect calorimetry measures oxygen consumption and can be used to estimate resting energy expenditure (REE), which increases in children with chronic liver disease. Although non-invasive, it is a technically difficult procedure and may not be easily performed in uncooperative children [80].

The above predictive equations, however, appear to perform poorly in infants and young children with ESLD. Efforts should be made to guide energy provisions with the aid of indirect calorimetry, especially when malnutrition is already present [24].

Accurately monitoring the nutritional status of these patients is essential in early interventions to correct deficiencies, thus improving growth and reducing both morbidity and mortality.

Oral nutrition should always be encouraged whenever possible because it is physiologic, maintains gastrointestinal tract immunity and gut barrier integrity, and reduces bacterial overgrowth [41]. As discussed in detail above, ready-to-use commercial preparations for infants with chronic liver disease should contain higher calorie amounts, MCT fat contents of about 50%, and branched-chain amino acids. Powder formulations allow one to customize the nutritional contributions in relation to the patient’s needs. More concentrated formulae containing maltodextrin may be useful for improving carbohydrate intake [25,53].

A commercially available and nutritionally complete powdered infant milk for the dietary management of acute and chronic liver disease is composed of dried glucose syrup, 49% MCT, soya and canola oil, whey and casein, vitamins, minerals and trace elements. The formula contains a higher-than-average content of branched-chain amino acids (30% of total protein) and lower-than-average sodium content (0.56 mmol/100 mL) than any other infant formula. The low lactose content, avoiding the saturation of the intestinal lactase capacity, is also an advantage of this formula when it is necessary to administer high amounts to increase the caloric input (e.g., by enteral feeding). There are no available published studies on this product, other than one report on its safety and tolerability, and there was no evidence of increased growth parameters when cholestatic children were given the product as their primary energy source [84].

Oral feeding is preferable only if sufficient energy and nutrient supply can be secured. In infants, except in some cases such as those due to galactosemia, breastfeeding should be encouraged for its numerous advantages, provided that supplementation with a breast milk fortifier containing MCT and proteins assures adequate intake of calories and nutrients. Later in life, children with CCLD require a complete nutritional assessment and nutritional therapy. Again, besides some rare inborn errors of metabolism (IEM) that may need special treatment, a standardized, hypercaloric diet with MCT should be used for preventing and/or correcting malnutrition. Small and frequent feeding may be useful to reduce anorexia and early satiety, prevent hypoglycemia, and avoid muscle catabolism.

When oral nutrition does not guarantee an adequate nutritional intake and causes poor growth because of intolerance to a large intake via the oral route, thus necessitating moderate calorie and high protein intake, patients should start nutritional supplementation. In this case, constant attention to palatability, osmolarity, or exceedingly high lipid levels is necessary When rapid growth is needed (e.g., before liver transplantation), enteral feeding with a nasogastric tube may be the best option. Bolus feeding is preferred in the first instance because it is physiological. When bolus feeding is not tolerated, it is appropriate to use continuous infusion administered by a peristaltic pump for up to 20 h/day [22]. One study, comparing an MCT-fortified formula administered orally versus by enteral nutrition (EN), showed that EN prevented malnutrition and growth impairment in infants with BA waiting for a liver transplant [46].

Night-time nasogastric tube feeding is another option; the child is allowed to eat ad libitum during the day and receives extra energy when sleeping, with the chance therefore of maintaining some normal feeding behavior [38]. Importantly, regular non-nutritive sucking and oral stimulation should be implemented to reduce the development of oral aversion. Feeding aversion may be prevented, in part, by promoting daytime oral intake as well as encouraging children to experiment with various age- flavors and textures [41]. Last, but not least, the oral route has trophic action on the intestine and liver, and it protects against infections.

When nasogastric tube feeding is not feasible or inadequate for meeting caloric needs, jejunal tube feeding [85], or even an endoscopic gastrostomy if portal hypertension is mild [86,87,88] may be proposed if a multidisciplinary team can provide active follow-up and care for the child and the device is available. Gastrostomy feeding, however, should be avoided in children with CCLD because there is a risk of peritoneal infection and variceal bleeding (if portal hypertension is present) [36,89]. Newer approaches, such as low-profile gastrostomy, designed to avoid worsening existing portal hypertension and avoid peristomal variceal complications, remain scarcely documented in this category of pediatric patients [90].

Some children with severe liver disease may require parenteral nutrition (PN). These include children who cannot tolerate oral nutrition due to osmotic diarrhea or vomiting, and those with recurrent variceal bleeding. PN is also indicated when enteral feeding fails to realize growth targets. The use of PN in children with compensated liver disease follows standard principles, and if it is a short-term indication, it is not associated with hepatobiliary dysfunction or worsened cholestasis [91]. On the other hand, in children with decompensated liver disease (usually also awaiting liver transplantation), prolonged PN might lead to worsening liver disease (parenteral nutrition-associated liver disease (PNALD) or cholestasis (PNAC) [92]. The risk is generally higher in infants, especially if premature, and in those with associated extreme short gut [93]. To prevent the onset of PNALD or PNAC, it is advisable to monitor bilirubin and to administer, when possible, simultaneous enteral calories, even in small volumes (so-called “minimal enteral intake”). Enteral feeding, at least intermittently, is protective for liver function and may restore mucosal integrity, minimizing small intestinal bacterial overgrowth and promoting bile flow [92]. In addition, the administration of UDCA has also been shown to have a beneficial effect in PNAC in adults and children, and it may therefore be indicated in these patients [94,95]. The prophylactic use of UDCA for parenteral nutrition cholestasis is of unproven efficacy but acceptable. PN requires strict monitoring, as it may result in fluid and sodium overload and worsening ascites, and central line-associated bloodstream infections have also been reported [96]. PN with a partially or completely fish-oil-based lipid content may be advisable for halting and reversing liver disease [97].

Altogether, PN has pros and cons; it may significantly improve the nutritional status prior to LT, with a beneficial effect on the outcome [96,98], but in some cases, it may aggravate jaundice [92] and introduce a risk of fatal vascular access-related complications in an already fragile subject [99]. The home management of enteral/parenteral nutrition is possible after appropriate family education [100,101].

## 4. Special Diets in Some Common or Special IEM Causing Cholestasis

### 4.1. Tyrosinemia Type 1

Tyrosinemia type 1 (OMIM # 276700) is a rare autosomal recessive disorder of the tyrosine degradation pathway, resulting in the accumulation of toxic metabolites (succinyl acetone) that are substrates for fumarylacetoacetate hydrolase, i.e., the enzyme that is deficient. The clinical manifestations vary and include neonatal cholestasis with acute liver failure, hepatocellular carcinoma, growth retardation, renal dysfunction, and porphyria-like syndrome with neuropathy [102]. The current specific treatment consists of 2-(2 nitro-4-3 trifluoro-methylbenzoyl)-1,3-cyclohexanedione (NTBC) paired with a tyrosine- and phenylalanine-restricted diet Many commercial products containing tyrosine- and phenylalanine-free amino acid mixtures and supplemented with vitamins and minerals are available [103]. The aim of combined treatment (NTBC and diet) is to provide adequate nutrition that allows normal growth and development, keeping the tyrosine levels in the blood and tissues under control [104].

### 4.2. Galactosemia

Classic galactosemia (OMIM # 23040) is a rare inborn disorder of carbohydrate metabolism caused by a defect in the GALT gene encoding the galactose-1-phosphate uridyltransferase, resulting in an inability to metabolize galactose to glucose. If not treated with a lactose- and galactose-free diet, this condition can cause neonatal cholestasis, liver failure, kidney damage, sepsis, mental retardation, and death. A galactose-restricted diet should be promptly initiated using soy formulas, not only in symptomatic infants but also in those with a highly suspicious newborn screening result [105]. With the introduction of solid foods, nutritional issues become more complicated, as trace amounts of hidden galactose are naturally found in fruits, vegetables, bread, and legumes [106]. The treatment guidelines for galactosemia, published in 2017, recommend eliminating sources of lactose and galactose from dairy products but allowing any amount and type of fruit, legume, vegetable, or mature cheese (with galactose contents <25 mg/100 g) in the diet, as they only have small amounts of galactose [107]. The restriction of dairy products is associated with inadequate calcium intake and with a risk of diminished bone mineral density, unless proper supplementation is initiated [108,109].

### 4.3. Hereditary Fructose Intolerance

Hereditary fructose intolerance (OMIM# 229600) is a disorder that arises in infancy, at the start of weaning, when fructose and sucrose are introduced into the diet, or even before if an infant is given medications containing sucrose. Generally, the child presents with recurrent vomiting, abdominal pain, (sometimes fatal) hypoglycemia, growth retardation, liver failure, and renal tubulopathy (in the case of long-term fructose exposure). Older patients who survive infancy develop a natural avoidance of sweets and fruits [110].>

### 4.4. Citrin Deficiency

Citrin deficiency, or neonatal-onset type II citrullinemia (OMIM # 605814), is a genetic disorder caused by a homozygous or compound heterozygous mutation in the SLC25A13 gene (603859) causing various metabolic abnormalities. It may present different clinical manifestations during the course of infancy to adulthood, ranging from cholestasis, fatty liver, and growth retardation in infancy, to liver dysfunction and neuropsychiatric symptoms in childhood and adulthood. Although in almost all infants it is a self-limiting condition, with symptoms and biochemical markers improving with age, the occurrence of severe hepatic dysfunction requiring liver transplantation has also been described [111,112]. It is recommended to maintain a low carbohydrate intake and a protein- and fat-rich diet with a protein–fat–carbohydrate ratio of 15–25%:40–50%:30–40% [113]. In fact, due to the impairment of NADH shuttling and glucose metabolism, patients have an aversion to carbohydrates and a peculiar preference for high-protein and high-fat foods, in contrast to patients with other urea cycle defects [114]. Excessive carbohydrate intake causes different symptoms such as fatigue, anorexia, weight loss, psychiatric symptoms, and liver failure, which is the most significant risk factor for adult-onset type II citrullinemia [113,115]. In patients with hypergalactosemia, the use of lactose-free milk containing MCT is associated with improved energy metabolism and liver function [116]. The administration of sodium pyruvate (0.1–0.3 g/kg/day) [83] improves clinical symptoms and helps in weight recovery [117,118].

## 5. Pre- and Post-Transplant Nutritional Status of Children with End-Stage Cholestatic Liver Disease

Although some causes of neonatal cholestasis have no specific treatment, affected children may benefit from appropriate nutritional support to prevent malnutrition and to correct macro/micronutrient deficiencies. This is paramount because a better pre-transplant nutritional status is associated with better post-transplant outcomes, and lower mortality and morbidity [18,68,119,120,121]. The nutritional needs of children with liver disease are outlined schematically in Figure 5.

A comparison of nutritional needs of children with chronic cholestatic liver diseases before and after liver transplantation [6,41,54,120] is shown in Table 3.

Useful literature on the nutritional management of children undergoing liver transplantation is limited. There are no pediatric studies showing which approach to feeding is superior (e.g., nasogastric, oral, or nasojejunal) and what impact it has on the post-liver transplantation outcomes. However, if the child eats orally up until the transplant, enteral feeding should be encouraged with a healthy, age-appropriate diet and also trying foods that they disliked before the transplant, as taste changes are common. Post-transplant feeding should start as soon as possible, ideally during the first 72 h if the child is stable [54]. After liver transplantation, weight gain seems to recover completely, despite previous malnutrition. Even mid-arm muscle mass and mid-arm fat start to rapidly improve within 3–6 months [41,120]. Although marked catch-up growth is observed in those who are more stunted before transplantation, transplanted children generally achieve final heights below their genetic targets [120,122]. Rejection, re-transplantation, and metabolic disease are independent risk factors associated with shorter stature. The use of steroids post-transplant for autoimmune (e.g., de novo autoimmune hepatitis) or allergic complications also contributes [123,124]. However, underlying genetic conditions (e.g., Alagille syndrome) may contribute to permanent low stature.

## 6. Post-Transplant Obesity with Fatty Liver and MetS Risk: The Malnutrition in Excess paradox

Most children catch up to their peers in weight by one-year post-transplant [125,126,127], but it may take up to five years to catch up in height [93]. This imbalance may lead to overweight and obesity within two years post-transplant. Careful nutritional counselling and close follow-up for the identification of children at risk for persistent overweight/obesity may help with targeted interventions to prevent not only obesity [128] but also long-term obesity-related comorbidities, such as metabolic syndrome [129,130] and non-alcoholic fatty liver disease (NAFLD), which has now been more correctly renamed metabolic-associated fatty liver disease (MAFLD) [131].

Post-transplant metabolic syndrome (PTMS) is increasingly recognized as a significant contributor to long-term morbidity and even mortality after solid-organ transplantation [132,133]. However, data on the prevalence of PTMS in children are scarce. Hypertension and glucose intolerance seem to be most common in the early post-transplant period. However, even five to ten years post-transplant, these comorbidities are much more common in children after liver transplantation than in the general population [134]. Liver transplant recipients are predisposed to these comorbidities predominantly due to the side effects of the immunosuppressive therapy used to prevent rejection. The side effects of glucocorticoids (GCs) are widely documented. GCs cause glucose intolerance by reducing peripheral insulin sensitivity (particularly in the muscles, the liver, and adipose tissue) and decreasing glucose utilization, and inducing dyslipidemia, hypertension, and hepatic changes with steatosis [135,136,137]. The high prevalence of impaired glucose tolerance (IGT) in liver transplant recipients is also due to the prolonged use of calcineurin inhibitors (CNIs). Tacrolimus is more diabetogenic than cyclosporine [138], as it impairs beta cell function [139] and/or stimulates insulin resistance [140]. CNI exposure is also associated with systolic hypertension owing to its direct nephrotoxicity, enhancement of sodium reabsorption, and induction of systemic vasoconstriction [141,142], and with increased circulating LDL cholesterol [143]. PTMS, influenced by weight gain and drug-induced hyperlipidemia and diabetes, is a crucial factor associated with the development of de novo NAFLD in liver transplant recipients [144]. Guidelines from the American Association for the Study of Liver Disease and the American Society of Transplantation recommend annual screening for obesity, hypertension, dyslipidemia, and diabetes mellitus with physical examination and fasting blood tests [145,146]. For this reason, it is essential to identify and test interventions to improve the screening and management of PTMS in order to improve long-term outcomes.>

### Sarcopenia

Sarcopenia is a reduction in muscle mass/muscle wasting. Few studies have evaluated its impact in this special pediatric population. Reduced muscle mass may be studied using several methods requiring cooperation to stay motionless during measurements, which is challenging for infants and young children. The tests include dual-energy X-ray absorptiometry (DEXA), CT, MRI, and bioelectric impedance analysis (BIA). The risks from radiation exposure restrict the use of CT, if not included in clinical staging (e.g., before OLT), while radiation-free MRI is expensive and may require narcosis in pediatric-age subjects. DEXA is a useful substitute, but whether all-body skeletal muscle mass and appendicular lean mass performs equally well, as markers of sarcopenia are controversial. The poor availability of age- and gender-matched normative data is a relevant obstacle.

CCLD pediatric patients may develop liver insufficiency and need early liver transplantation. As in adults, both malnutrition and sarcopenia compromise pediatric post-transplantation outcomes [147,148]. In liver-transplanted children, sarcopenia is associated with relevant clinical outcomes (growth retardation, length of hospitalization, and rate of readmission) [149]. The use of an internal comparison for the diagnosis of sarcopenia may cause different series to have different cutoffs, which hampers the generalization of results [150,151].

Myopenia, caused by decreased protein synthesis and increased protein degradation, is another clinical manifestation of end-stage liver disease (ESLD). It is associated with adverse clinical outcomes, confirming the importance of rehabilitation strategies [152].

Although several valid malnutrition screening tools with which to identify nutritional risk are available, an increased risk of protein malnutrition may be masked by edema, and further work regarding tools with which to distinguish between protein–energy malnutrition (PEM), sarcopenia, and cachexia are required for the pediatric age group. Functional assessment of muscle sarcopenia in pediatric CCLD is not generally performed because appropriate muscle function tests have not been developed for early childhood, and a standardized assessment of muscle function for the diagnosis of sarcopenia in young pediatric patients is currently lacking [150,151].

CT-derived body metrics (e.g., the skeletal muscle index (SMI), psoas muscle index (PMI), psoas muscle surface area (PMSA), and subcutaneous fat index (ScFI)) are measurable components of sarcopenia, frailty, and nutrition. The relationship between preoperative metrics and post-LT outcomes in pediatric recipients < 1 year-old with cirrhotic liver disease is unknown. Fragmentary data suggest that the preoperative values are significantly reduced compared with controls, and the values are correlated with moderate to severe postoperative infections and with longer hospital and ICU stays [153]. A higher re-operation rate and longer hospital stays following transplantation, but not waitlist mortality, were associated with lower PMI [154] or psoas muscle surface area (PMSA) [155]. Sarcopenia, evaluated by total psoas muscle area (tPMA) measurements from computed tomography (CT) imaging, was prevalent in patients with pediatric ESLD awaiting LT, highlighting the need for nutritional support before LT and/or after LT in the PICU [156]. Only one study [150] examined both muscle mass and muscle function in children with CLD, as recommended by the recent EWGSO consensus [157].

## 7. Conclusions

The causes of malnutrition in CCLD are varied. Although some causes of cholestasis have no specific treatment, all children with CCLD should have a periodic evaluation of clinical, anthropometric, biochemical, and instrumental nutritional indicators. In particular, it is important to identify specific nutritional needs and any deficiencies that require early preventive or corrective treatment. Supplemental nutrition, including MCT, essential fatty acids, branched-chain amino acids, and vitamins, is crucial for overcoming anorexia and preventing growth retardation.

Malnutrition is associated with increased risks of morbidity and mortality, and it also affects the outcomes of liver transplantation and long-term survival, so that failure of nutritional care is an indication for reviewing liver transplant timing. To improve life expectancy and quality of life, patients with CCLD need careful assessment of their nutritional statuses by a multidisciplinary team capable of initiating appropriate interventions. The complex dietary management for this type of malnourished patient requires new studies to improve commercially available and nutritionally complete infant milk formulae.

## Figures and Tables

**Figure 1 nutrients-13-02785-f001:**
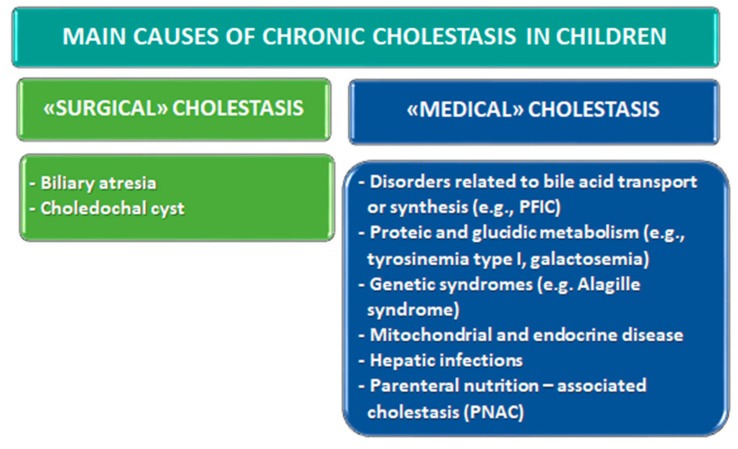
Main etiologies of prevalently extrahepatic (“surgical”) or intrahepatic (“medical”) pediatric chronic cholestatic liver disease. HFI, hereditary fructose intolerance; PFIC, progressive familial intrahepatic cholestasis.

**Figure 2 nutrients-13-02785-f002:**
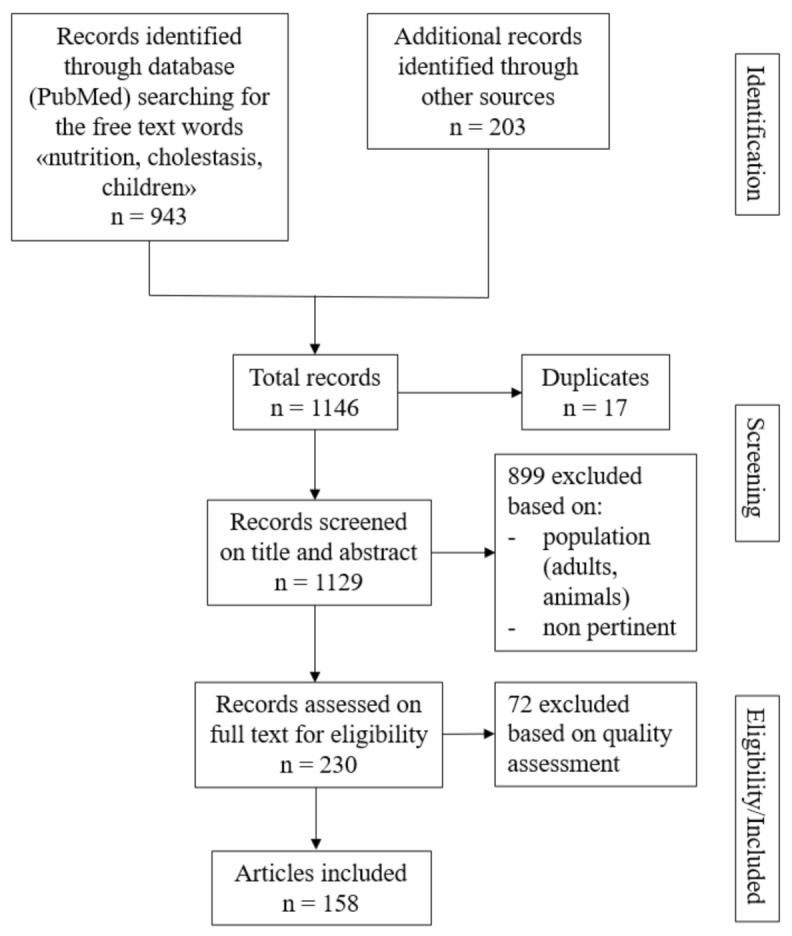
Flow chart showing the selection process to identify studies included in the article.

**Figure 3 nutrients-13-02785-f003:**
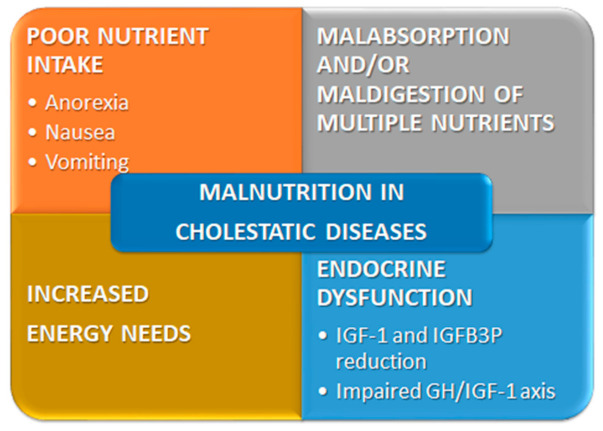
Main factors that determine malnutrition in pediatric chronic cholestatic liver disease. GH, growth hormone; IGFB3P, insulin-like growth factor binding protein 3; IGF-1, insulin-like growth factor-1.

**Figure 4 nutrients-13-02785-f004:**
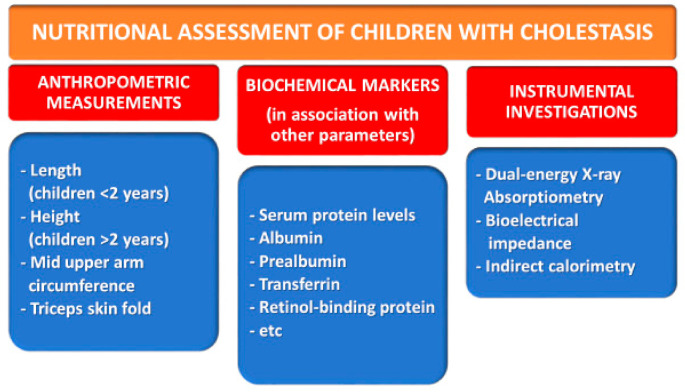
Nutritional assessment of pediatric chronic cholestatic liver diseases.

**Figure 5 nutrients-13-02785-f005:**
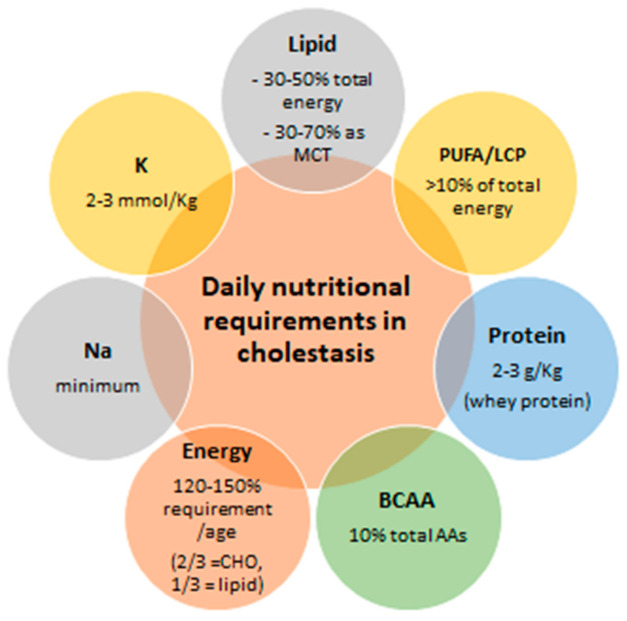
Nutritional needs of children with liver disease. BCAA, branched-chain amino acids; CHO, carbohydrates; LCP, long chain polyunsaturated fatty acids; MCT, medium-chain triglycerides; PUFA, polyunsaturated fatty acids.

**Table 1 nutrients-13-02785-t001:** Weight-based equation for calculating energy requirements (kcal/day) according to the FAO/WHO/UNO [23].

Age	Males	Females
3–10 years	22.7 (Weight) + 495	22.5 (Weight) + 499
10–18 years	17.5 (Weight) + 651	12.2 (Weight) + 746

**Table 2 nutrients-13-02785-t002:** Synopsis of fat-soluble vitamin deficiencies and toxicities.

Vit	Signs and Symptoms of Deficiency	How to Monitor	Supplementation	Toxicity
**A**	Dry skin Xerophthalmia Night blindness.	Plasma retinol/retinol binding protein molar ratio > 0.8 relative dose response When serum retinol levels < 20 µg/dl, RDR test is indicative of Vit A deficiency when the plasma retinol concentration increases after exogenous administration of Vit A dose.	3000–10,000 IU/day <10 kg: 5000 UI/day, Oral >10 kg: 10,000 UI/day, Oral 50,000 UI/1–3 monthly, IM	Hepatic and neurologic toxicity Development of long bone fractures
**D**	Hypocalcemia/hypophosphatemia/tetany Osteomalacia and rickets History of reduced intake, decreased cutaneous synthesis, altered absorption/impaired metabolism in the liver (i.e., where Vit D2 and D3 undergo 25-hydroxylation). Phenobarbital treatment. Breastfeeding	Serum 25-OH-D (Vit D deficiency < 20 ng/mL; insufficiency < 30 ng/mL) Ca, P, AP, PTH, Bone radiography/Bone densitometry used to identify osteomalacia, osteopenia or rickets	Cholecalciferol: 800–5000 IU/day, Oral 1.25-OH cholecalciferol: 0.05–0.2 µg/kg/day, Oral	Hypercalcemia leading to depression of the central nervous system and ectopic calcification. Hypercalciuria leading to nephrocalcinosis
**E**	Hypo- or a-reflexia Ataxia Impaired vibratory sensation Proximal muscle weakness Ophthalmoplegia Degenerative lesions of the retina Irreversible neurological lesions if Vit E deficiency remains untreated	Vit E/total lipids ratio (increased lipoprotein levels in cholestasis may falsely elevate serum Vit E levels in a patient with Vit E deficiency) Vit E deficiency: <0.6 mg/g (age <1 year) <0.8 mg/g (age >1 year)	Alpha-tocopherol acetate: 15–25 to 25–200 UI/kg/day, Oral TPGS (tocopheryl polyethylene glycol-1000 succinate): 15–25 UI/Kg, Oral	Potentiation of Vit K deficiency coagulopathy Diarrhea Hyperosmolality (TPGS)
**K**	Hemorrhagic disease (other risks of bleeding: portal hypertension gastrointestinal bleeds, thrombocytopenia, platelet dysfunction, reduced hepatic synthesis of other coagulation factors	Prothrombin time International normalized ratio Protein induced in Vit K absence II (PIVKA II) <3 ng/mL Deficiency can be diagnosed if these values improve after a dose of parenteral vitamin K	2.5–5.0 mg/day from twice a week to every day 5–10 kg: 5 mg, oral >10 kg: 10 mg, oral 5–10 mg/day every two weeks, IM	Hemolytic anemia in glucose 6-phosphate dehydrogenase-deficient infants

AP, alkaline phosphatase; IM, intramuscular; PTH, parathyroid hormone; Vit, vitamin.

**Table 3 nutrients-13-02785-t003:** Nutritional needs of children with chronic cholestatic liver diseases before and after liver transplantation.

Before Liver Transplant	After Liver Transplant
**Energy intake** 130–150% EAR	**Energy intake** 120% EAR
**Carbohydrates (40% to 60% of total energy)** 15–20 g/kg/day as monomers, polymers, and starch Balancing the hypoglycemia from end-stage liver disease and hyperglycemia from insulin resistance	**Carbohydrates** 6–8 g/kg/day as monomers, polymers, and starch Warning: Consider the diabetogenic potential of tacrolimus when it is used for immunosuppression
**Proteins (9% of total energy)** 3–4 g/kg/day BCAA-enriched formula can be used (10% of total amino acid) Low protein-diet is needed only when severe encephalopathy is present. Once encephalopathy is resolved, the patient should resume a diet with appropriate protein supply because long-term restriction <2 g/kg/day can induce endogenous muscle protein consumption	**Proteins** 2.5–3 g/kg/day
**Fats (40% of total energy; 10% of which as LCPUFA)** 8 g/kg/day with 30–50% as MCTs. Warning: MCT contents >80% without adequate supplementation of PUFA can lead to a deficiency of essential fatty acids	**Fats** 5–6 g/kg/day After liver transplantation, when bile flow is established and malabsorption is resolved, children fed with high MCT-containing supplementation pre-transplant can transition to standard formula
**Fluids and electrolytes** Fluid requirement is normal for actual weight, unless restriction is needed because of ascites or edema. Sodium intake is 1 mmol/kg/day and potassium about 2 mmol/kg/day	**Fluids and electrolytes** A “no added salt” diet (3 g sodium/day) is recommended to prevent water retention associated with steroid therapy

BCAA, branched-chain amino acids; EAR, estimated average requirement; LCPUFA, long-chain polyunsaturated fatty acids; MCT, medium-chain triglycerides; PUFA, polyunsaturated fatty acids. Bold characters indicate the different nutrients’ groups. Modified from Yang et al. (2017) [41].

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
