# Peer review of "Malnutrition in Pediatric Chronic Cholestatic Disease: An Up-to-Date Overview"

_nutrients, 2021, doi:10.3390/nu13082785_

Round 1

Reviewer 1 Report

General:

  • This is a very comprehensive review of the literature, both recent and relevant older literature.
  • It provides a great summary of factors that contribute to malnutrition in pediatric chronic cholestatic liver disease, before and after transplant
  • Please edit grammatical issues (for example, several areas in which a coma or parenthesis is included in a sentence inappropriately)
  • This manuscript is very long and I think there are areas which could be edited to make for a more succinct read

Abstract

  • Recommend shortening the abstract and simplify language in abstract. For example, erase “what became evident when reviewing the literature was…” to simply say “Malnutrition in children with chronic liver disease is multifactorial with multiple potential nutritional deficiencies”
  • This sentence is confusing: “Solutions available in everyday clinical practice, in general and in conditions characterized by specific etiology, are summarized.” Recommend “Solutions available for the clinical management of children with liver disease, in general, as well as those directed to specific etiologies, are summarized.”

  1. Introduction
  • Recommend using newer terminology “choledochal malformation” as opposed to choledochal cyst
  • Biliary atresia and choledochal cyst (choledochal malformations) can both affect the intra-hepatic bile ducts so I recommend against this classification as “intrahepatic” vs “extrahepatic” and don’t think Figure 1 is completely necessary
  • Consider revising this concept, “Malnutrition is mainly but not only- the result of an intensely reduced bile-acid-dependent absorption of fats and fat-soluble nutrients”—it’s difficult to quantify to what degree cholestasis versus fibrosis/cirrhosis contribute to malnutrition (for example in a child with BA and adequate bile flow following Kasai, they may still have growth failure)

  1. Causes of Malnutrition
  • It’s unclear why the treatment of pruritus is included under causes of malnutrition, specifically “poor nutrient intake” (recommend removing this information)
  • Under “Increased energy needs”: To my knowledge it is not clear that activation of thyroid hormone by bile acids is primarily responsible for a hypermetabolic state in children with cholestatic liver disease (and it doesn’t seem that the 2 citations referenced actually support this statement)
  • Be sure to clarify that protein restriction is almost never indicated in children with CCLD
  • “Water and electrolytes”: this section makes it seem like you should sodium restrict all infants with CCLD, which I would not necessarily agree with. I would make it clear that 1 mg/kg/day of sodium is a starting point for infants but some may need more. I agree that in children with CCLD and cirrhosis, should not try to correct the serum sodium as it may drive ascites. However, would only recommend sodium restriction in children who have developed ascites.
  • I recommend shortening each of these sections and ensuring you are including practical / clinically relevant information. For example, there is a paragraph on testing for PUFA and EFA deficiency—in what children would you recommend testing? Certainly not all children with CCLD need regular fatty acid profile testing.
  • Make sure citations are included where appropriate (for example, this sentence needs citation: “Decreased levels of 25-OH (vitamin D) are present in more than half of cholestatic patients, and this is positively correlated with serum calcium. Decreased BMD was pre-sent in more than half of the studied cholestatic patients and was correlated with low serum calcium rather than vitamin D levels.”)
  • Recommend using <20 ng/mL to define vitamin D deficiency

  1. Nutritional management
  • This section is well-written and high-yield. The impact will only be increased by shortening / editing the sections which precede it!
  • It’s not clear if the authors are suggesting Heparon Junior over other infant formulas in infants with CCLD. I just want to ensure there is no conflict of interest for the authors with this product, given this is the only specific product mentioned (I suspect not, since no conflicts of interest were disclosed).

  1. Special diets
  • Recommend correcting the galactosemia section—it is due to a defect in the GALT gene, which encodes the protein galactose-1-phosphate uridyltransferase (as opposed to saying that it is “a defect in the galactose-1-phosphate uridyltransferase gene”)
  • HFI section: onset can also happen if an infant is given medications containing sucrose (ie can happen before weaning). Also, this section doesn’t mention anything about the dietary management of HFI, which it seems is the point of the section.

  1. Conclusions
  • Both the introduction and conclusion of the manuscript comment on “educational training regarding nutritional guidelines for stakeholders, and improving family nutritional health literacy” but these concepts aren’t really included in the body of the manuscript. I would suggest removing these from the Introduction and Conclusion unless they are specifically addressed in the body of paper.

Reviewer 2 Report

The manuscript by Tessitore et al. Causes, management, and assessment of malnutrition in pediatric chronic cholestatic disease: an up-to-date overview is a comprehensive review of 158 papers published during the last 20 years. The authors have analyzed all aspects of malnutrition in those clinical conditions and provided clear information on possible treatment regimens.

The study is well done, succinctly written and nicely illustrated, informative, and clinically important.

There is nothing to correct.

Reviewer 3 Report

Minor points like title, captions and other minor inconsistencies are attached in the revised manuscript.

Round 2

Reviewer 1 Report

The manuscript is significantly improved in it's current form. Thank you to the authors for their thorough response.